# Light microscopy image segmentation using active contours driven by local image information for environmentally friendly fired-clay bricks design and characterization

Aurel Mihail Baloi[1,2,3], Mihaela Streza[4], Bogdan Belean [2,4]*

**1** Interdisciplinary Center for Advanced Studies (CISA-ICUB), University of Bucharest, Bucharest, Romania, **2** Graphit Innovation Factory, Drobeta-Turnu Severin, Romania, **3** Faculty of Administration and Business, University of Bucharest, Bucharest, Romania, **4** National Institute for Research and Development of Isotopic and Molecular Technologies, Center for Advanced Research and Technologies for Alternative Energy, Cluj-Napoca, Romania

* bogdan.belean@itim-cj.ro

## Abstract

This study proposes an innovative approach for the recycling of dry wastewater sludge in the production of fired clay bricks, aiming to create environmentally friendly and cost-effective building materials. Consequently, the proposed research focuses on optimizing the thermal and mechanical properties of ceramic bricks, while incorporating sewage sludge as a pore-forming agent in brick manufacturing process. For the assessment of porosity in ceramic brick mixtures with varying volumetric proportions of ash, clay and wastewater sludge, a novel method for microscopy image analysis is proposed. This advanced image segmentation method uses active contours driven by local image information to accurately estimate brick porosity from light microscopy images, coping with light reflections, intensity inhomogeneity, and the defocus effect present on the sample surface. The thermal properties of the ceramic blocks were assessed using non-contact infrared lock-in thermography and thermogravimetric analysis (TGA). The obtained results indicate that incorporating sewage sludge increases porosity, thereby improving the material's insulating properties, with the cost of reduced mechanical strength. Nevertheless, the compressive strength for a mixture containing 15% of sewage sludge meets the criteria for the production of first-class bricks, according to ASTM C62 standards for brick manufacturing. The present study demonstrates that incorporating sewage sludge into fired clay ceramic bricks production is a feasible and sustainable approach, offering significant advantages for waste management and promoting energy-efficient building design.

**Data availability statement:** All relevant data are within the manuscript and its Supporting Information files. Moreover, high-quality images for figures from 1 to 7 were uploaded, for the rest of figures, they are already made available within the document.

**Funding:** The research was funded by a grant from the Romanian Ministry of Education and Research, UEFISCDI, under project no. 61TE, code PN-IV-P2-2.1-TE-2023-0300, and by the Experimental-Demonstrative Project PN-IV-P7-7.1-PED-2024-2349, grant number 32PED/08/01/2025, project acronym MLCELLM.

**Competing interests:** The authors have declared that no competing interests exist.

## 1. Introduction

Due to the growing interest in energy reduction in buildings, the production and optimization of ceramic blocks incorporating waste materials offer a promising approach to making the ceramic sebctor more cost-effective and environmentally friendly. Simultaneously, evaluating building materials for energy efficiency is a critical factor in promoting economic and environmental development worldwide. Most building energy consumption is directed toward achieving thermal comfort, with air conditioning loads increasing as heat transfer into buildings rises. Efficient materials can drastically reduce energy consumption, lower the operational costs, and mitigate environmental impact through decreased greenhouse gas emissions. As emphasized in [1], the choice of materials that improve thermal insulation can enhance building performance, reduce dependency on energy-intensive cooling systems, and contribute to sustainable development goals. The amount of energy consumed by materials production and the heat transfer through the building envelopes are parameters that influence the thermal comfort, the production costs and the environmental impact. The thermal conductivity is the most important parameter for a high-quality thermal insulating material. In this regard, it is extremely important to improve the insulating properties of ceramic blocks by reducing the thermal conductivity and the density of the raw materials. However, this process often results in a lower mechanical strength which can be problematic in terms of building stability. For this reason, an appropriate compromise between thermal and mechanical properties of ceramic blocks must be achieved by manufacturers.

Over the past few years, many groups published their research findings regarding the advantages and challenges of using different waste materials and pore formers for improving the thermophysical properties of the ceramic building materials. The incorporation of paper sludge in ceramic blocks and its technical and environmental benefits were reported in [2] and [3] with acceptable mechanical, thermal and acoustical properties. The addition of recycled waste ceramics and paper pulp in fired clay bricks reduces thermal conductivity, but increases permeability and decreases compressive strength, as underlined in [4]. Similar pore formers (i.e., waste micro cellulosic fiber) improve the thermal performance of fired clay bricks [5]. It was shown in [6] that using Bayer process bauxite waste and agricultural residues as additives in brick production leads to lower thermal conductivity and increased porosity, resulting in higher thermal insulation. Considering large scale manufacturing, the addition of slag and zeolites, coal or organic residues were analyzed in [7–9]. Incorporating bamboo leaf as an additive for fired clay brick leads to increased porosity on samples during the firing process and hence reducing density and thermal conductivity [10]. Additionally, studies such as [11] highlight the incorporation of waste materials, like solid mine waste, into construction products to enhance properties and promote sustainability.

Due to industrialization, population growth, and rising quality standards for water effluents, sewage sludge production from wastewater treatment (WWT) has increased significantly and continues to grow. Consequently, the treatment of sewage sludge is gaining considerable attention. For instance, [12] proposes a practical

framework for the digital transformation of sewage treatment plants to automate processes, improve decision-making, reduce costs, and enhance operational efficiency. Additionally, wastewater sludge has been identified as a promising feedstock for sustainable biodiesel production. In this context, Shayan and Saman [13] designed a supply chain network for large-scale biodiesel production from fermented wastewater sludge. Approximately 30% of sewage sludge produced in the European Union is recycled into agriculture by incorporating it into agricultural soils, which improves soil fertility and crop yields without exceeding heavy metal concentration limits, making it a viable fertilizer option [14]. Several studies have explored the valorization of wastewater sludge in building materials, noting both advantages and challenges related to thermophysical properties. Incorporating wastewater sludge into building materials generally increases porosity, which can enhance thermal insulation but may reduce mechanical strength [15]. Research in [16] demonstrates that sewage sludge can be used as a raw material for eco-cement, bricks, ceramic materials, and lightweight aggregates, with performance comparable to traditional Portland cement. Incorporating sewage sludge into clay mixtures is highly feasible for producing extruded ceramic bricks, particularly with optimal moisture content [17]. However, adding sludge to fired clay bricks increases porosity while also increasing shrinkage during the drying process and reducing strength properties. A strategy proposed in [18] shows promise for fabricating high-performance lightweight ceramisites from municipal solid sludge and fly ash wastes. The environmental impact of incorporating this waste into fired clay bricks is generally low, with heavy metal concentrations within standard limits in most studies and gas emission analyses showing minimal air pollution [19,20]. Additionally, the calorific value of the dry waste fraction can save 10% to 40% of fuel, offering further advantages in energy efficiency [21].

Despite existing studies, comprehensive global data on waste streams remain limited, with only partial exceptions. For instance, in the United States, approximately 13.84 million metric tons per year (MT/y) of dry solid sludge are produced, half of which is not beneficially utilized, as detailed in [22]. Moreover, large-scale recycling solutions for dry solid sludge from municipal wastewater remain unresolved, making the incorporation of wastewater sewage sludge into brick manufacturing an emerging area of interest and a viable waste management alternative. This study leverages the expertise of the Cemacom Zalău Structural Clay Manufacturing Plant, incorporating wastewater solids from the Aquaserv Târgu-Mureș Municipal Water Authority as a partial substitute in the composition of ceramic blocks, which are traditionally produced using raw materials such as clay and ash. Sample bricks were produced in collaboration with local industry, ensuring adherence to industry standards and quality control measures. As previously highlighted, incorporating dry sewage sludge into red ceramics increases porosity, enhancing insulation properties, and provides an efficient waste management solution. This aligns with broader efforts to improve aquatic environmental quality through environmentally friendly recycling strategies [23] and the growing trend of investing in creative industries in Romania's capital and other administrative regions [24].

The manufacturing process for red ceramic bricks is presented in this paper, with further analyses performed on the resulting brick units to validate this potential sludge disposal solution. Porosity is estimated using image processing techniques applied to microscopic images of red ceramics, while thermal properties are determined through specific experimental setups. Additionally, thermogravimetric curves are used to assess mineral weight loss due to thermal decomposition during firing procedures. When selecting the most suitable microscopy imaging technique, image resolution is a critical factor, requiring a balance between cost and sample preparation effort. In this case, light microscopy is advantageous for its simplicity and cost-effectiveness, particularly when micrometer-scale resolution is sufficient for features such as porosity estimation. The light microscopy image segmentation procedures suitable for porosity estimation are discussed further, alongside the proposed approach.

## 1.1. Light microscopy image segmentation

Image processing plays a critical role in visualizing and interpreting structures across various imaging modalities, from light microscopy to atomic-scale techniques. Prior to image registration, these modalities rely on computer vision-based

segmentation to extract structural information. The primary goal is to identify meaningful features corresponding to image objects within the image under analysis. In medical image analysis for example, segmentation plays an important role in tasks such as visualization, measurement and reconstruction of shapes and volumes [25–27] or even in image guided-surgery [28]. Mohammed et al. [29] propose a classification of the existing segmentation techniques applied for medical image segmentation. Additionally, significant research effort has been devoted to developing image segmentation approaches for material characterization [30]. The main image segmentation approaches can be broadly classified as active contour models (ACMs), clustering procedures, and machine learning (ML) approaches. Clustering-based techniques identifies image objects and features by classifying pixel intensity values in homogenous clusters while ignoring the spatial information, which make them sensitive to image artifacts. Considering segmentation efficiency among the clustering-based algorithms, fuzzy C-means (FCM) has been widely used for image segmentation [31], whereas special attention is given to fast convergence due to their iterative nature [32]. Considering the machine learning approaches for image segmentation, both supervised and unsupervised ones are available. The supervised ML-based segmentation benefit of an increased accuracy, mainly due to the fact that input data is labelled and well known, as shown in [33,34]. On the other hand, unsupervised learning has the advantage of automatic segmentation without any prior knowledge of the object features within the training dataset [35], but the price of an increased computational complexity is to be taken into account as underlined in [36]. Thus, machine learning approaches that account for both global and local information are proposed in [37] for pixel level segmentation. Alongside machine learning, improved level-set segmentation approaches are available, such as a no-weight initialization level-set model combined with bilateral filtering [38,39]. Inspired by these approaches, but aiming to reduce the computational complexity, we propose a region-based image segmentation model guided by local image information, to address the specific requirements of optical microscopy image segmentation. A two-level fusion approach for construction waste classification based on image processing is also proposed in [40].

In region-based image segmentation, initially the gradient information has been used to guide an evolving contour toward the object boundaries. Reduced accuracy is achieved in the presence of weak and noisy edges. Active contour models improve segmentation accuracy by formulating an energy function based on information such as intensity, color, or texture to evolve an initial contour toward object boundaries [41]. These approaches are applied to porosity estimation in ceramic materials, which is crucial for understanding their mechanical and thermal properties. The ceramic blocks are sliced to obtain flat surfaces, whereas high-resolution images of these surfaces are acquired using light microscopy techniques. Further on, the resulted images are processed for porosity estimation. Considering the quantitative evaluation of convolutional neural networks, U-net neural networks and level set procedures detailed in [42], along with the increased complexity of the neural network approaches as referred to the segmentation task, the paper proposes an adaptive level-set based segmentation procedure for porosity estimation in the acquired optical images. Machine learning approaches are not necessarily suitable for this task due to their increased complexity compared to the requirements (i.e., determination of porous regions based on pixel intensity values). Nevertheless, the envisaged segmentation task is not trivial considering phenomena such as light reflections on the sample surface, intensity inhomogeneity and the defocus effect specific to light microscopy. The main drawback of the defocus effect is that it leads to a depth of view relatively shallow, especially at higher magnifications. While objects or structures within the focal plane appear sharp and clear, showing fine details and well-defined edges, the structures below the focal plane appear blurred compared with the focal plane ones. This causes clustering procedures and classic active contour models approaches to deliver inaccurate segmentation results due to weak and missing edges caused by the defocus effect. Considering the pore size as the projection of the pore in the focal plane, the aforementioned structures bellow the focal plane interfere with the pore surface, reducing pores estimated size. More than that, light reflexions on marginal areas below the focal plane may lead even to increased intensity compared to overall luminance image profile. To overcome the previous disadvantages such as partial identification of ceramic pores, we proposed an adaptive segmentation approach based on a curve evolution process guided by the minimization of an energy functional. It is well-known that some traditional level set methods used either the

image gradient [43] or the global information [44] to drive the evolving curve(s) towards the true boundaries. Additional terms in curve evolution known as regularization terms are added to minimize the length of the contour or the area of the region enclosed by the contour. A curvature based term is used in our case. Nevertheless, accurate segmentation of the defocus areas as pore regions can not be obtained, especially in cases where structures below the focal plane are characterized by increased pixel intensities compared to the background. To cope with the defocus effect, we introduce a local energy term within the classical level-set method. Consequently the energy functional is composed of three parts, global term, local term and regularization term. The local term is computed considering the extended narrow band of the evolving curve overlapped over the Laplacian of the original image. In the context of the defocus areas, the Laplacian operator is particularly useful because blurred regions exhibit smoother intensity transitions, leading to lower values of the Laplacian. This local blur information is used for accurate pore segmentation, and therefore accurate porosity estimation is achieved.

## 1.2. Highlights and contributions

The primary objective of this study is to propose a novel approach for recycling dry wastewater sludge in clay brick production in an environmentally safe and efficient manner, providing a cost-effective alternative for brick manufacturing. The brick manufacturing process is carefully monitored, and the thermophysical properties of building ceramics are analyzed to ensure enhanced efficiency and quality. A comprehensive approach combining computational design and engineering is employed. Thermal properties are determined using non-contact infrared lock-in thermography, providing precise evaluation of the material's response to thermal stimuli. Thermogravimetric analysis (TGA) is used to study mineral weight loss during thermal decomposition. Porosity, the most critical parameter, is analyzed through a curve evolution approach specifically designed to address the limitations of traditional light microscopy techniques.

The paper is structured as follows: after the introductory section, the preparation of ceramics from raw materials and the research methodology, including investigation techniques, are detailed in the Materials and Research Methodology section. The Results and Discussion section presents the thermal parameters, thermogravimetric analysis, and physico-mechanical parameters of the red ceramic samples. A brief synthesis of the results is provided at the end, along with an outlook for producing sustainable and cost-effective building materials.

## 2. Materials and research methodology

### 2.1. Materials preparation

The following raw materials were used for sample preparation: a mixture of yellow and grey clay (the basic mixture), ash and dry sewage sludge as pore-forming agent. The sewage sludge was added in volumetric percentages of 5%, 10% 12% and 15% to the basic mixture. The starting materials were previously dried in the oven at 90 °C for 16 hours, then were grounded and further used to prepare homogeneous mixtures having the composition presented in Table 1. In order to facilitate the modeling of the samples, around 20 wt. % water was added to the prepared mixtures. The homogeneous mixtures were added in layers in a specially manufactured hardened steel mold and pressed using a hydraulic press (1.96 MPa). Finally, samples having the dimensions of 80 x 40 x 15 (mm) were obtained. The samples thus obtained were dried

Table 1. Sample preparation.

| Sample | Basic mixture (%vol) | Ash (%vol) | Sewage sludge |
| --- | --- | --- | --- |
| CB | 85 | 15 | 0 |
| N17 | 80 | 15 | 5 |
| N18 | 75 | 15 | 10 |
| N19 | 73 | 15 | 12 |
| N20 | 70 | 15 | 15 |

in the oven at 90 °C for about 12 hours, and furthermore, were thermally treated in a Nabertherm calcination furnace at 900 °C for 5 hours. The temperature heating rate was 2°C/min. After the heating treatment, the samples were maintained in the oven and were slowly cooled to ambient temperature with no forced ventilation, with a cooling rate of 40°C per hour, up to 200°C. The final cooling phase was carried out at a controlled rate of approximately 10 °C per hour, allowing the specimens to gradually reach ambient temperature without inducing thermal stress. The resulting ceramic blocks were cut into 15 mm x 15 mm x 15 mm cubes and their thermophysical properties were investigated.

## 2.2. Physico-mechanical properties

Compressive strength, density and water absorption (by mass) were obtained by standard methods applied on the resulting cubes. The compressive strength was determined on Controls Advantest 9 hydraulic press with a load rate of 0.2 MPa/s, and the accuracy of the force recording of 0.01MPa. The morphology of the resulting pores was investigated by image processing on the optical images of sections of the resulting ceramics. The geometry of the porous regions, their size and distribution together with the mineralogical composition define the thermal performance of novel building materials. Consequently, accurate identification of porous region is important and it is performed by means of light microscopy and segmentation procedures on the acquired images in our study.

**Porous region identification.** The areas of interest on the registered microscopic images, corresponding to the pores within the ceramic blocks, can be defined as the projection of the pores in the focal plane. These porous regions are delineated by the so-called background, which in this case, is represented by the actual ceramic brick material. Relative to the focal plane, the pore size can be defined as the projection of the pore in the focal plane. The structures found below the focal plane appeared blured, whereas structures within the focal plane appear sharp and clear, showing fine details and well-defined edges. Moreover, light reflections on marginal areas below the focal plane may lead even to increased intensity compared to overall luminance image profile. This phenomena, known as the defocus effect, poses significant challenges in image analysis, as it complicates the accurate detection and segmentation of features due to the weak and missing edges in the blurred regions. To address these issues, the Laplacian operator can be employed to highlight the areas affected by the defocus effect. By transforming the image data, the transformation by Laplacian operator helps in identifying the locations where the defocus effect is present, including the areas where light reflections in the marginal areas below the focal plane may lead to an increase in intensity compared to the overall luminance profile of the image.

To overcome the disadvantages posed by the defocus effect, such as the partial identification of ceramic pores, an adaptive segmentation approach is proposed. This approach utilizes a curve evolution process guided by the minimization of an energy functional from eq. (1), which is geoverned by three main terms, global term $E^G$, local term $E^L$ and regularization term $E^R$. The proposed method is inspired from the well-known Chan-Vese region-based segmentation model (CV) [45], a particular case of Mumford-Shah (MS) active contour model [46], which incorporates similar global and regularization terms. The local term $E^L$ was added to account for the defocus areas in the pore segmentation process, aiming accurate porosity estimation. An example of such defocus areas underlined by the use of the Laplacian operator applyed on the original image are illustrated in Fig 1. The overall energy functional terms are detailed on the subsequent sections.

$$E = \alpha E^G + \beta E^L + E^R \tag{1}$$

*Global term*

Let the microscopy image be denoted by the two-dimensional array of intensities $u_0$ with $u_0 : \Omega \to R$, where $\Omega$ is a bounded open subset of $R^2$. The Mumford–Shah introduce an energy functional defined in eq. (2), used for finding the pair *(u,C)* for a given image $u_0$, where *u* is an approximation of $u_0$ and *C* is the closed segmenting curve for image objects. Minimizing the energy functional over u and C leads to the determination of closed contours which perform image segmentation.

$$E^{MS}(u, C) = \int_\Omega |u_0(x, y) - u(x, y)|^2 \, dxdy + \mu \int_\Omega |\nabla u(x, y)| \, dxdy + \gamma Length(C) \tag{2}$$

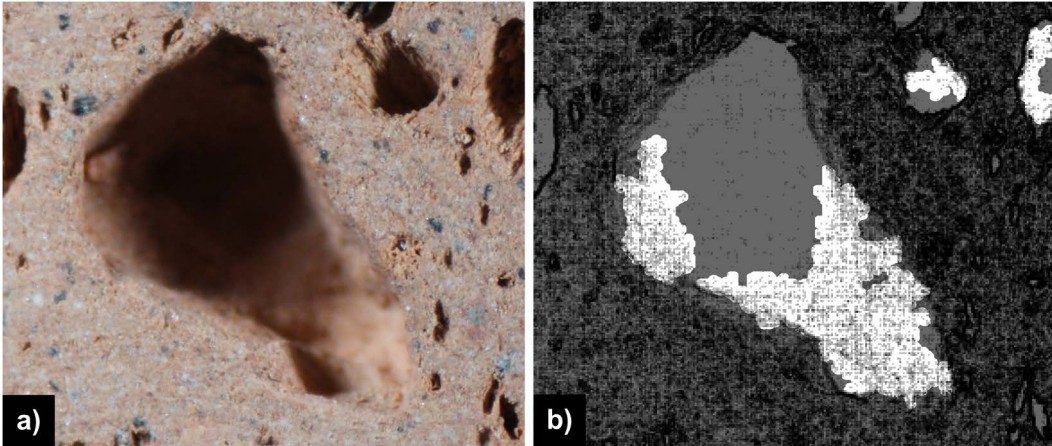

**Fig 1. a)** section of the original image sample ID 1095 including pore with expressed defocus areas; **b)** defocus areas underline by the Laplacian operator.

Similar solution for finding these closed segmentation contours is the Chan-Vese model, where the following energy functional is minimized:

$$E^{CV}(c1, c2, C) = \mu Length(C) + \lambda_1 \int_{\text{Inside(C)}} |u_0(x,\ y) - c1|^2\, dxdy + \lambda_2 \int_{\text{Outside(C)}} |u_0(x,\ y) - c2|^2\, dxdy$$

(3)

Within the aforementioned energy functional, the $\lambda_1$, $\lambda_2$ and $\mu$ are positive constants ($\lambda_1 = \lambda_2 = 1$) and $c1$ and $c2$ are the intensity averages of $u_0$ inside $C$ and outside $C$, respectively.

The global term $E^G$ is derived from the CV model and it is defined as follows:

$$E^G(c1, c2, C) = F_1(C) + F_2(C) == \int_{\text{Inside(C)}} |u_0(x,\ y) - c1|^2\, dxdy + \int_{\text{Outside(C)}} |u_0(x,\ y) - c2|^2\, dxdy$$

(4)

The Chan–Vese(CV) model replaces the curve $C$ by the level set function $\phi(x, y)$, with positive values for the point *(x,y)* inside $C$, negative values for *(x,y)* outside the curve $C$ and $\phi(x, y) = 0$ if the (x,y) point is placed on the $C$ curve. Consequently, the global term can be written as:

$$E^G(c1, c2, \phi) = \int_{\Omega} |u_0(x,\ y) - c1|^2\, H(\phi(x,y)) dxdy + \int_{\Omega} |u_0(x,\ y) - c2|^2\, (1 - H(\phi(x,y))\, dxdy$$

(5)

where $H(z)$ is the Heaviside function.

To sum up, the main procedure is defining an initial contour of the level set which delineates an inside and an outside region of the curve. The means of the aforementioned regions are computed, whereas the difference of the two means forms the global term of the level set energy functional. The level set energy is minimized, leading to a small difference between pixel intensity within the same region.

*Regularization term*

In order to assure a smooth level set, we add a regularization term *L(C)* which is defined with respect to the evolving curve C. Let us consider C as a smooth closed planar curve

$C(p) : [0, 1] \to \Omega$, with parameter $p \in [0, 1]$. The length regularization term can be written as: $L(C) = \oint_C dp$. By replacing the curve C with the level set $\phi(x, y)$, the length regularization term can be reformulated as:

$$L(\phi = 0) = \int_\Omega \left|\nabla H(\phi(x, y))\right| dxdy = \int_\Omega \delta(\phi(x, y)) \left|\nabla \phi(x, y)\right| dxdy \tag{6}$$

where $\delta(z)$ is the Dirac delta function.

The use of the length regularization term makes sure that the evolving curve which minimizs the overall energy functional is as short as possible and avoids the occurrence of small and isolated curves in the final result.

*Local term*

First we define an extended narrow band around the curve C as $B_d(C) = \{x \epsilon \Omega : dist(x, C) < d\}$, where *dist(x,C)* is the distance froma point *x* to the curve *C*. The local term is introduced in (7) and uses the local information given by the Laplacian operator applied on the original image to improve the segmantationa capability in case of images characterized by the local defocus effect.

$$E^L(d1, d2, C) = \int_{\text{Inside } B_d(C)} \left|c_k * \nabla^2 u_0(x, y) - d1\right|^2 dxdy + \int_{\text{Outside } B_d(C)} \left|c_k * \nabla^2 u_0(x, y) - d2\right|^2 dxdy \tag{7}$$

The d1 and d2 are the average values of the *original image Laplacian* $\nabla^2 u_0(x, y)$ inside and outside the extended narrow band $B_d(C)$,whereas $c_k$ is an average convolutional operator, with $k < d/2$. In our case, where the presence of defocus areas are targeted to be part of the segmented object areas, it is significant to analyze each pixel with respect to its local neighborhood. The analysis is performed with respect to the image Laplacian illustrated in Fig 1. Thus, the increased local average of the image Laplacian compared with the corresponding narrowband average leads to the dealiniation of areas with reduce variation (Fig 1b) as part of the segmentaed object areas.

Similar to the global term, the local term $E_L$ is formulated using the level set function $\phi(x, y)$, which leads to the following expression:

$$E^L(d1, d2, \phi) = \int_\Omega \left|c_k * \nabla^2 u_0(x, y) - d1\right|^2 H(\phi(x, y))dxdy + \int_\Omega \left|c_k * \nabla^2 u_0(x, y) - d2\right|^2 (1 - H(\phi(x, y))dxdy$$

Considering the previously defined terms, the overall energy functional is described as follows:

$$E^T(c1, c2, d1, d2, \phi) = \alpha E^G(c1, c2, \phi) + \beta E^L(d1, d2, \phi) + E^R(\phi) \tag{8}$$

where $\alpha$ and $\beta$ underline the influence of the global and local term onto the curve evolution. Results of the overall segmentation process are illustrated in Fig 2, which shows the luminance profile of the original optical image of the thin section from Fig 1, where the darker areas represent the solid phase, whereas the bright zones represent the pore phase Fig 2a. In the same figure, the closed contours determines by the curve evolution process underline the pore areas, which are marked using the binary mask from Fig 2b overlayed on the original image.

## 2.3. Thermogravimetric analysis (TGA)

The thermogravimetric (TG) curves are recorded to study the mineral weight loss due to thermal decomposition. Thermogravimetric experiments were conducted with an SDT Q600 instrument (TA Intstruments), using the row material mixture in a Pt crucible. SDT Q600 allows simultaneous measurement of mass variation (TGA) and differential heat exchange (DSC) of the same sample from the room temperature up to 1200 °C. It is equipped with a dual balance with automatic

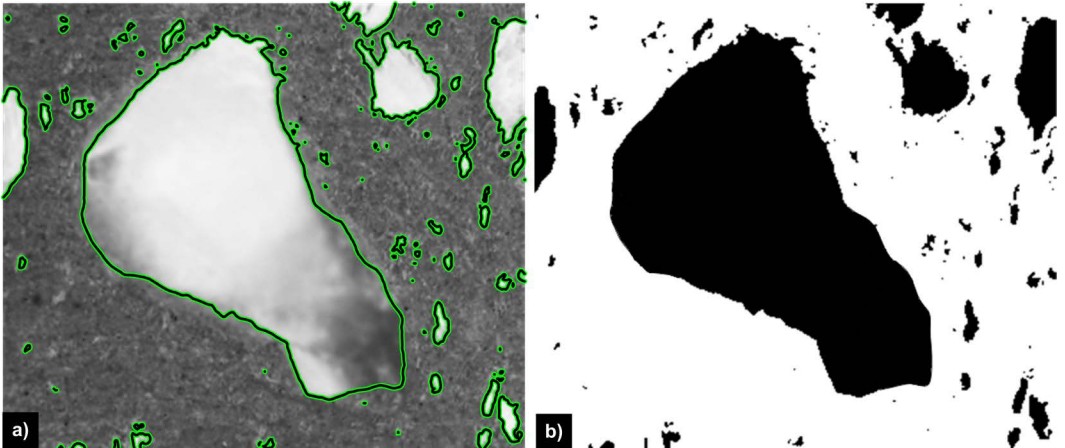

**Fig 2. a) Results of the proposed level set procedure in case of the image ID together with b) the segmentation mask which marks the detected porous structures.**

compensation of shaft dilatation (sensitivity of 0.1 μg). The DSC thermal flow data is dynamically normalized using the instantaneous weight of the sample at a given temperature. The temperature measurement is done with 2 Platinum/Platinum-Rhodium thermocouples, providing a direct measurement of the sample temperature, the reference temperature and the differential temperature. The thermal measurements were carried-out in argon atmosphere, after a pre-calibration procedure. The furnace heating speed was set at 10°C/min. The accuracy of the calorimetric measurements is 2%.

### 2.4. Thermal properties

The thermal effusivity of the samples was obtained by using the PPE technique in the front configuration (FPPE), while non-contact infrared lock-in thermography technique (LIT) was used to get their thermal diffusivity, as previously was done for other type of building materials [4]. Knowing the values of the thermal effusivity and thermal diffusivity, the thermal conductivity was calculated according to the following equation: $k = e\,(\alpha)^{1/2}$, where α is the thermal diffusivity, e is the thermal effusivity and k is the thermal conductivity of the investigated sample. A few points concerning the basic theoretical and experimental aspects will be resumed here.

In the front detection configuration (FPPE), for thermally thick sensor and sample, the thermal effusivity of the sample can be measured by performing a frequency scan of the phase of the PPE signal, according to Eq. (9):

$$tan\Theta = \frac{(1 + R_{mp})\,exp(-x)\,sin(x)}{1 - (1 + R_{mp})\,exp(-x)\,cos(x)}$$

(9)

where $R_{mp}$ represents the reflection coefficient of the thermal wave at the interface material (m)- sensor (p), $L_p$ is the thickness of the sensor, and $a$ is the reverse of the thermal diffusion length ($a = 1/\mu$), with $x = a_p L_p$ and $\frac{e_m}{e_p} = \frac{1 + R_{mp}}{1 - R_{mp}}$

The thermal diffusion length is $\mu = (\alpha/\pi f)^{1/2}$ with $f$ the modulation frequency.

To meet the theoretical requirements, an opaque $LiTaO_3$ crystal (provided with CrAu electrodes) with a thickness of 500μm and a thermal diffusivity of $\alpha = 1.1 \times 10^{-6}$ m²/s was used as pyroelectric sensor. A blue laser diode (P = 300m, l = 450nm) was used as an excitation source. The modulation frequency of the diode was changed in the frequency range 1,25 Hz. A very thin layer of silicon grease was used as coupling fluid between the sample and the sensor, in order to improve the thermal contact. The laser beam was targeted to the sensor and expanded by using a mirror and a divergent lens.

The thermal diffusivity of solid samples by using the LIT technique can be calculated from the shift of the phase (which is a time delay in the propagation of thermal wave as compared to a reference signal). The phase shift $Dj$ has the following expression:

$$\Delta\phi = -\sqrt{\frac{\pi f}{\alpha}}x = ax$$

(10)

where x is the distance from the punctual heat source, $a = 1/\mu$ is the slope of the phase versus distance plot and $f$ is the excitation frequency. At a big distance from the punctual heat source, the thermal wave can be approximated by a plane wave and accordingly, the thermal diffusivity can thus be calculated according to Eq. 10.

The experimental IR setup for measuring the thermal diffusivity included a heat source (Laser Quantum OPUS, with l = 532 nm and P @ 2W), a waveform generator, an infrared camera (FLIR 7200 series, with a 256 x 320 pixel array of [1.5,5.1μm] sensitive InSb quantum detectors) and a computer for data acquisition. The optical axis of the camera was set perpendicular to the surface of the investigated sample. The signals delivered by the infrared camera and the reference frequency ($f_0 = 2$ Hz) were sent to the lock-in embedded detection module, which outputs the continuous component image (f = 0) as well as the amplitude and phase images of the $f_0$ – component to a PC.

## 3. Results and discussion

Considering the mixtures detailed in Table 1, batches of 80x40x15 (mm) ceramic bricks were obtained for each mixture, according to the technological process described in section 2.1. Samples of 15x15x15 (mm) size from each batch were investigated in terms of their thermo-physical characteristics using the research methodology described in section 2.

### 3.1. Physico-mechanical properties

Mercury intrusion porosimetry (MIP) together with the proposed level-set approach and a typical image processing procedure were used to estimate the porosity in case of samples from each batch. The final porosity values $m_p$, $IA_{ls}$ and $IA_{th}$ included in Table 2 are obtained as the mean of the porosity measurements in case of multiple samples drawn from each batch, coreponding to N18 – N20 mixtures. For exemplification, optical images for three samples from batches N18, N19 and N20, together with the binary images underlining the detected pores using the proposed active contours approach driven by local image information are illustrated in Fig 3.

In order to maintain linear margins and achieve uniformity across all images, the acquired images by means of optical microscopy were cropped, leading to 3500x3240, 3500x2689 and 3500x3185 pixel sizes scaled images, corresponding to the analysed samples drawn from N18, N19 and N20 batches, respectively. This preliminary process excludes also dark areas located on the edges of the optical image where pores can not be distinguished. As referred to MIP techniques, which applies pressure to force mercury into the pores, it is well know that it reliable measure porosity for pore sizes ranging from about 3 nanometers (nm) to 1 milimeter (mm). For this reason, the population of pores in case of the

**Table 2. Physico-mechanical properties of mixtures.**

| Clay mix batch | $f_{c,med}(P)$ [N/mm²] | $\rho_{med}$ [g/cm³] | $a_i$ [%] | mp [%] | $m_p$ stdev | $IA_{ls}$ [%] | $IA_{ls}$ stdev | $IA_{th}$ [%] | $IA_{th}$ stdev |
|---|---|---|---|---|---|---|---|---|---|
| Control brick | 47 | 1.73 | 4 | 4.2 | | – | | – | |
| N17 | 42 | 1.68 | 5.2 | 6.8 | 0.81 | 4.8 | 0.82 | 4.1 | 0.79 |
| N18 | 34 | 1.63 | 10.4 | 11.2 | 0.65 | 9.3 | 0.72 | 8.7 | 0.71 |
| N19 | 28 | 1.58 | 12.3 | 13.5 | 0.64 | 11.2 | 0.82 | 10.4 | 0.69 |
| N20 | 26 | 1.56 | 13.7 | 14.2 | 0.59 | 12.0 | 0.69 | 11.5 | 0.72 |

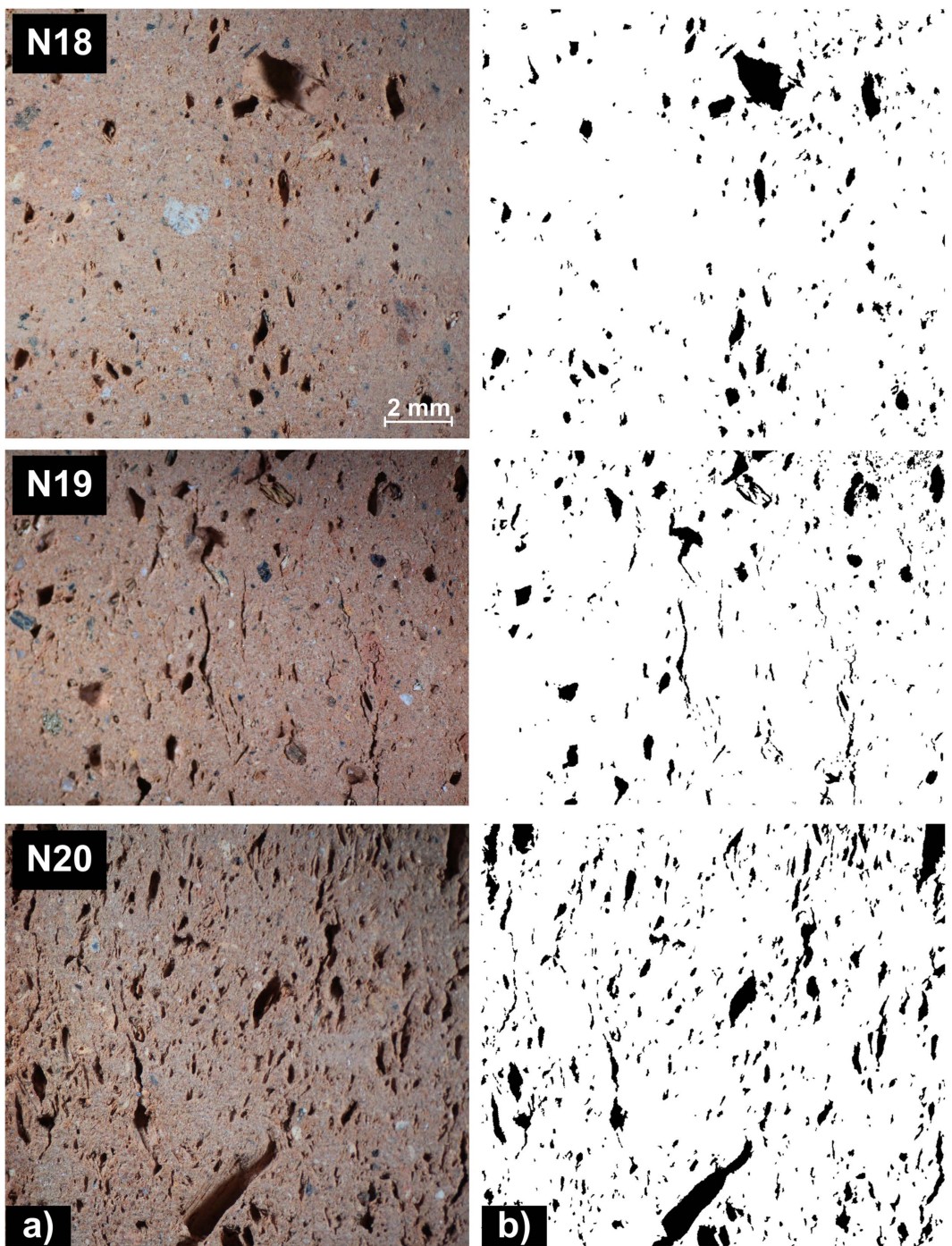

**Fig 3. Optical mages of samples from batch N18, N19 and N20 together with the binary image representing pore distributions obtain by the proposed active contours approach driven by local image information.**

samples from Fig 3, is characterized by separate histograms for pores under the 0.7 mm$^2$ threshold (micropores), and also for macropores larger then the considered threshold value. Thus, the histograms presented in Fig 4 illustrate the distribution of both micro- (Fig 4a) and macro- pores (Fig 4b) sizes within the samples under analysis, in terms of their frequency (number of pores).

The first two peaks Fig 4a indicate a high frequency of smaller pores, which is consistent with the fine-grained nature of the base material. The uniform distribution of pores, coupled with their reduced size, is expected to enhance the compressive strength of the material. This is based on the premise that smaller and more evenly distributed pores minimize stress concentration points within the structure, thereby reducing the likelihood of crack initiation and propagation under load. Moreover, Fig 3 shows that pores have a somewhat elongated eliptical shape which can confer to sample a good behavior to environmental agents (thaw-freezing cycle, chemical attacks, etc).

Despite the inherent limitations of mercury intrusion porosimetry (MIP), such as its restricted capability to measure very large pores accurately, the pore size distribution of our samples, which falls within the operational range of MIP, justifies an objective comparison between the two approaches for porosity measurement. Let us consider $m_p$ as the porosity of sample brick measured using MIP techniques and $IA_{ls}$ and $IA_{th}$ the porousity expressed in percents, measured by the proposed image processing approach and also by classical image processing procedure. Considering an addition of up to 15% sewage sludge is introduced into the sample mixture, Table 2 shows that porosity ($m_p$) increases from 4% to 13.2% in case of MIP measurements, whereas image processing analysis show an increase of up to 12% in case of the proposed approach which accounts for the defocus effect. It is to be mentioned that, typical image processing procedure, which involves segmentation based on closed contours guided by thresholding procedure, yield results that deviate more from those obtained through mercury intrusion porosimetry. For the same N17 to N20 mixtures, Table 2 also illustrates variation of both average density and water absorption measurements of the samples, denoted by $\rho_{med}$ and $a_i$, respectively. The density ($\rho_{med}$) was reduced by aproximately 10%, while the water absorption ($a_i$) increases from 4 to 13.7%. The calculated mean for the measured water absorption coefficient of 13.7% falls below the maximum allowable limit of 17% specified by ASTM C62 for fired clay bricks intended for use in areas subject to severe weathering, indicating compliance with the standard and suitability for such environmental conditions.

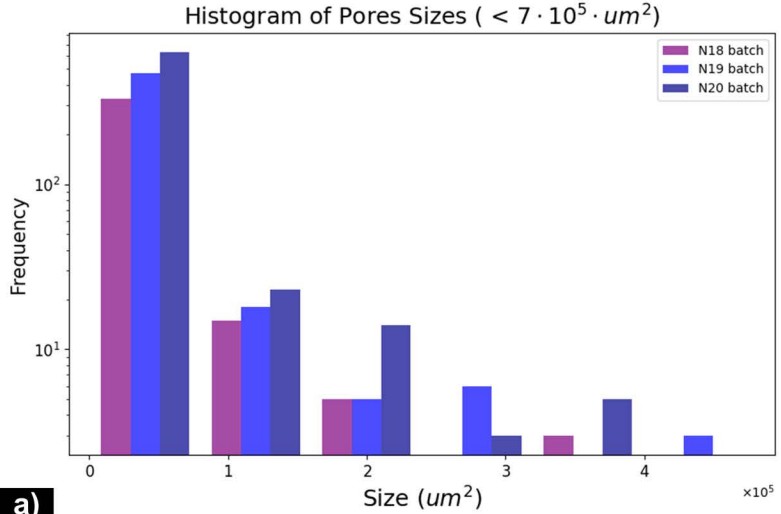
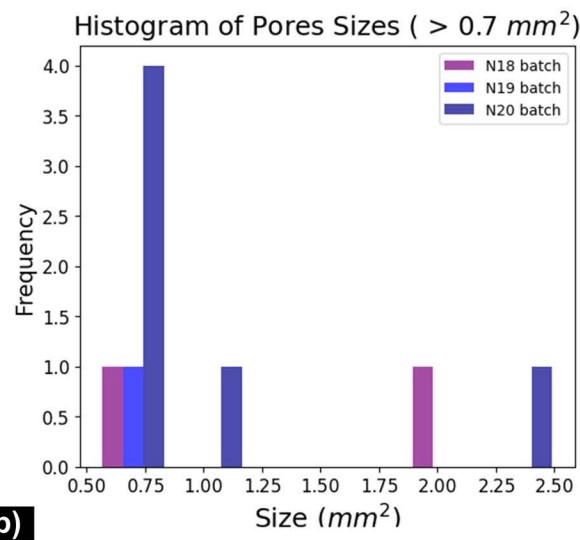

**Fig 4. Distribution of pores having their size a) lower and b) higher than the 0.7 mm$^2$ in case of the N18, N19, N20 samples.**

The mercury porosity values $m_p$ are slightly higher, than the $a_i$ ones, mainly because the MIP technique detect a wider range of pore sizes, including smaller pores that water might not penetrate due to surface tension or connectivity issues. On the other hand, water absorption predominantly measures larger, connected pores. Consequently, the large increase in water abosbtion is mainly due to the the increase of connected pores.

The Fig 5 illustrates the relationship between porosity and defocused regions in image sections extracted from N20 samples, which include large pores areas. In the upper row, images labeled a), b), and c) depict the original sections of the material, showcasing visible pore structures. The corresponding lower row contains binary masks labeled a'), b'), and c'), which highlight regions impacted by the defocus effect. Together, the original images and their respective masks offer a comparative view of the structural features and the defocus areas present in the samples. To provide a more in-depth analysis, quantitative data related to Fig 5 is presented, in terms of the percentage of regions affected by the defocus effect, the estimated porosity using the classical Chan–Vese active contour method, and the intersection between these two areas for each sample image. It is important to note that the porosity values computed using the classical approach do not necessarily reflect the overall porosity of the sample images, as the selected regions are concentrated around areas with large pores. Related to the relation between the defocus effect and the porous regions, in image 6.a, 8.74% of the area is affected by defocus, the estimated porosity is 18.19%, and the overlapping region between pores and defocus is 6.17%. For image 6.b, the defocus-affected area is 10.03%, porosity is estimated at 21.42%, and the overlap is 8.46%. In image 6.c, 5.5% of the area is defocused, porosity is 14.23%, and the intersection is 4.27%. The average overlap percentage between porosity and defocused regions is calculated based on the ratio between the respective values in each case. Notably, this overlap reaches approximately 75%, indicating that a substantial portion of the defocused areas

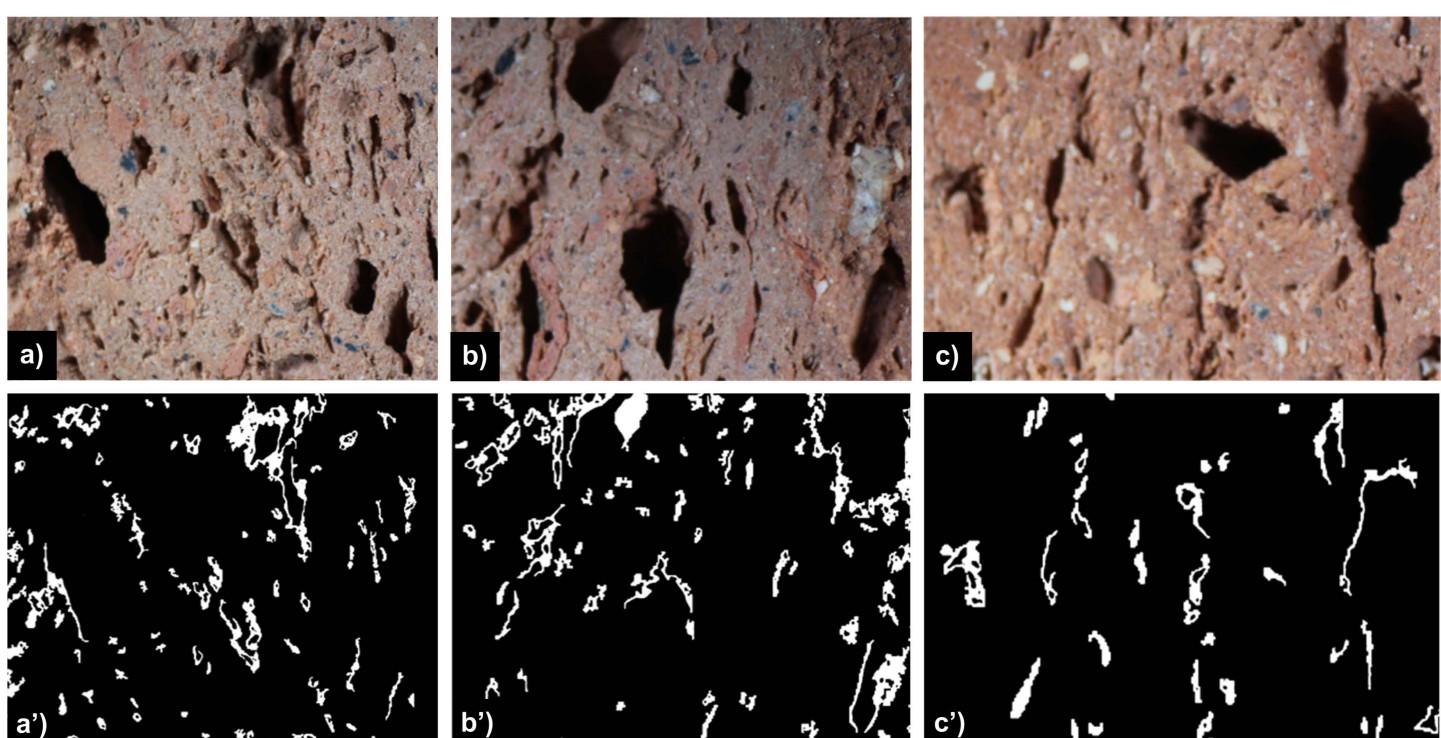

**Fig 5. Visualization of porosity and defocused regions in large pores sections from N20 samples. a) The upper row displays the original image sections a), b) and c) while the lower images row shows the corresponding binary masks highlighting areas affected by the defocus effect a'), b') and c').**

corresponds to pore regions. This percentage is in agreement with minor discrepancy observed between the porosity estimates obtained using the classical Chan–Vese method and those derived from the adaptive active contour approach.

The compressive strength of the samples was determined on three cubic specimens with dimensions of $15 \times 15 \times 15$ mm for each of the samples (N17, N18, N19, and N20), and their mean values was considered. Increasing total porosity by sewage sludge addition has the effect of reducing the average compressive strength of the specimens $f_{c,med}$, from 47N/mm$^2$ to 26 N/mm$^2$ (see Table 2), a normal decrease due to the reduction of the effective load bearing cross-section of the material. The obtained results are discussed in terms of the compressive strength of high-quality fired clay bricks reported in state-of-the-art literature. The compressive strength is significantly influenced by the incorporation of various additives, the firing temperature, and the material composition. As reported in [47], additives such as fly ash, glass cullet, and water treatment sludge can enhance the compressive strength, while higher firing temperatures generally lead to stronger bricks. Thus, bricks containing up to 70% water treatment sludge have better compressive strength values compared to those with only fired clay brick waste, reaching an average level of approximately 20 MPa. Aeslina et al. highlighted in [48] that incorporating 5% sewage sludge waste (SSW) in brick production yields bricks of acceptable quality, meeting the necessary standards for construction applications. In [49] Amir et al. utilized sewage sludge from Senggarang and Perwira in the fabrication of bricks intended for non-load-bearing applications, with results indicating that bricks incorporating Senggarang sludge outperformed those made with Perwira sludge. Moreover, in [50] was reported the production of fire clay bricks with 4% fly ash and glass cullet content which achieved a compressive strength of over 17.2 MPa, while maintaining low firing temperatures (i.e., 950 °C). In comparison, the brick specimens proposed in [51] incorporating regular iron wire and cast-iron powder demonstrated even higher compressive strengths, further reinforcing the potential of these waste materials to produce environmentally sustainable and mechanically robust bricks. The results presented in [52] indicate that lower replacement ratios (e.g., 20% or 40%) for clay with water treatment sludge may offer a balance between environmental benefits and mechanical performance, but as the proportion of WTS increases, the negative effects on compressive strength become more significant. The best results were obtained for the lowest sludge ratio (i.e., 20%) where the compresive strength reaches 16 MPa. As a result of our study, the obtained compressive strength of 26 MPa in case of the N20 samples meets the criteria for the production of first-class bricks, in accordance with ASTM C62 standards for brick manufacturing, and falls within the range of high-quality bricks reported in the literature.

To provide a deeper understanding of the material behavior and to enhance the practical relevance of the study, an in-depth analysis is introduced, aiming to establish a clear quantitative relationship between the bricks density, proportional with the porosity levels, and the corresponding compressive strength of the bricks. Fig 6 shows that the average compressive strength values of the specimens decreased progressively by approximately 10.6% (N18), 27.7% (N19), and 44.7% (N20) compared to the reference sample N17.

Using the Gibson-Ashby scaling law [53], it is possible to predict to a certain extent the behavior of porous materials relative to the variation in relative density:

$$P\prime = C_2 P_s \left( \frac{\rho\prime}{\rho_s} \right)^n$$

(11)

where $P\prime$ and $\rho\prime$ refers to the properties of the compressive strength and density, whereas the lower indices $s$ refers to the same properties, but for the reference sample, with no enhanced porosity. The coefficient $C_2$ is given by Gibson and Ashby for the compressive strength of metallic foams to lie typically between 0.1 and 1.0, with a theoretical exponent $n = 1.5$. In our case the coefficient $C_2$ is considered approximately 1, and the power exponent $n$ was found to be 5.57. This indicates a more rapid decay of compressive strength with increasing porosity compared to metallic foams (n = 1.5). This observation is consistent with the susceptibility of ceramic-based materials to crack initiation and propagation as the porosity increases. Nevertheless, the power exponent falls within the interval of 3.0 to 6.0, corresponding to dense

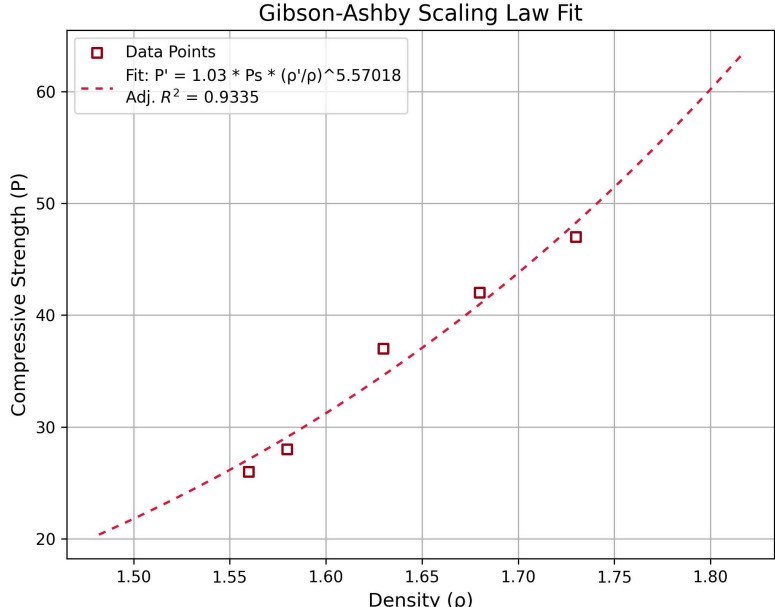

**Fig 6. Fitting of compressive strength data for fired clay bricks incorporating sewage sludge using the Gibson–Ashby scaling law.** Experimental compressive strength values are plotted as a function of sample density, and the fitted curve describes the relationship eq. (11) The fitting parameters, including the coefficient $C_2$, exponent n, and the adjusted coefficient of determination ($R^2$), are indicated on the plot.

fired clay bricks [53]. This suggests that, despite the increased porosity introduced by sewage sludge incorporation, the mechanical performance of the obtained bricks remains within acceptable limits for high-quality fired clay products, confirming their suitability for practical construction applications.

According to Table 2, the porosity evaluation of bricks incorporating increasing proportions of sewage sludge (5%, 10%, 12%, and 15%) was conducted using three techniques: classic level-set image segmentation ($IA_{th}$), adaptive level-set image segmentation ($IA_{ls}$), and mercury intrusion porosimetry ($m_p$). The results clearly reveal distinct trends in both porosity estimation and variability among these methods as ilustrated in Fig 7.

Classic image processing consistently produces lower porosity values with moderate variability; for example, in the 5% sludge group, $IA_{th}$ yields a mean porosity of 4.1% with a standard deviation of 0.79%. This method, while straightforward, is limited by its reliance on fixed thresholds and its inability to accurately capture pore structures that lie outside the focal plane, making it less effective in characterizing complex or heterogeneous porosity. In contrast, the adaptive level-set method yields higher porosity values, reflecting its enhanced capability to detect fine-scale and out-of-focus pore features through dynamic boundary adaptation. For instance, in the same 5% sludge group, $IA_{ls}$ reports a mean of 4.8%, closer to the mercury porosimetry value of 6.8%. Notably, $IA_{ls}$ exhibits slightly higher variability than the classic method, with a standard deviation of 0.82% compared to 0.79%. This pattern persists across all sludge contents, suggesting that the adaptive segmentation introduces minor variability, possibly due to its dynamic boundary detection, but also enhances structural accuracy. Mercury intrusion porosimetry ($m_p$), serving as the reference, consistently reports the highest porosity values with the lowest standard deviations. For example, at 15% sludge content, it yields a mean porosity of 14.2% with a low standard deviation of 0.59%. While adaptive level set segmentation introduces slightly more variability than the classic approach for the N17 to N19 samples, it provides porosity measurements that are significantly closer to those obtained via mercury intrusion porosimetry. This makes $IA_{ls}$ a valuable compromise between non-destructive image-based techniques and the high-accuracy, but invasive, mercury intrusion based method.

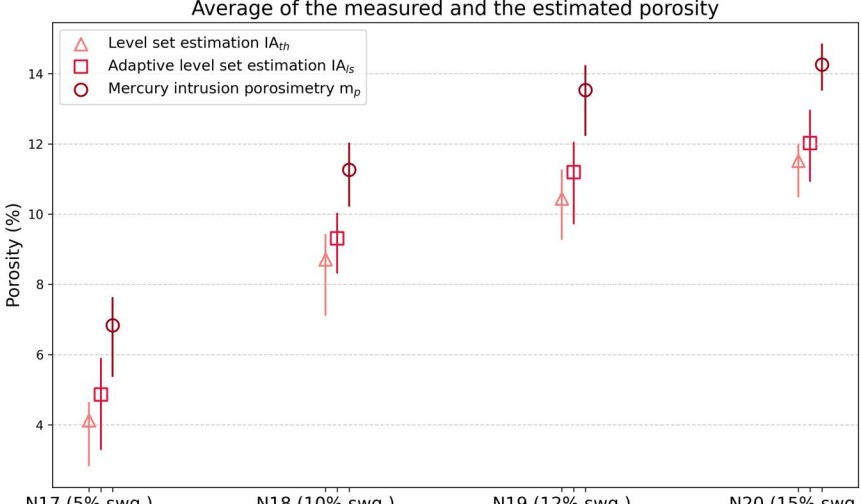

**Fig 7. Porosity evaluation for bricks with varying sewage sludge content (N17-5%, N18-10%, N19-12%, and N20-15%).** Each group presents porosity values derived from three techniques: classic level-set image analysis $IA_{th}$, adaptive level-set-based image analysis $IA_{ls}$, and mercury intrusion porosimetry $m_p$. For each measurement, the mean porosity is displayed as a central marker, with vertical lines indicating the minimum and maximum bounds.

For a more precise characterization of the brick fabrication process, the thermogravimetric analysis is performed considering the N20 samples. The subsequent section provides the thermogravimetric analysis results, addressing the total mass loss of the final mixture in relation to the estimated porosity.

### 3.2.  Thermal decomposition of raw materials by DTA-TG

Figs 8, 9 show the DTA-TG profiles for sewage sludge and basic mixture in argon atmosphere with a heating rate of 10°C/ min. This heating rate showed an optimal profile of the thermal decomposition process. TG curves show that with increasing temperature, the combustion samples takes place with an associated weight loss and heat release. Sewage sludge exhibits a quite complicated decomposition process, due to its complex chemical composition.

Based on the DTA results, four regions can be found:

The first region on TG curve (5.9%) is due to the moisture removing (drying) which is an endothermic process which is clearly observed on DTA curve (125°C). The second region (15.6%) appears due to removal of volatile organic compounds and it is accompanied by a low heat release (~380°C). The third stage of decomposition with the main weight loss (36%) due to the ignition and burning of the organic compounds is a highly exothermic process (~500°C). The last regions (2.3%) is due to the decomposition of inorganic compounds detected in sewage sludge. A weak, broad endothermic peak, is detected between 900°C and 1000°C. A total mass loss of 60% occurs for the dry sewage sludge, which confirms its potential use as pore former in ceramics. The total mass loss of the final mixture is around 16%. The final mixture undergoes basically a similar thermal behavior: removing water accompanied by a strong endothermic process (at 170°C), ignition and burning of ash and organic compounds accompanied by an exothermic peak at 480°C, and, in the last region, the decomposition of inorganic matter (calcite, caolinite) detected in basic mixture.

The differences between the stages of the decomposition process result from different amount of moisture, volatile, organic and inorganic matter (calcite, caolinite, quartz) detected in raw materials. This increase of the total mass loss of the final mixture can be attributed to the increased number of pores with a diameter below 2 microns, which are not measured by mercury porosimetry and which originate in the residues from the sludge burning.

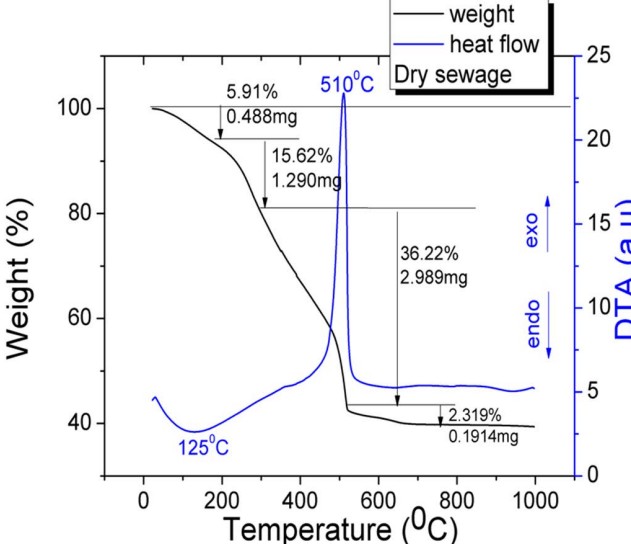

**Fig 8. TG for dry sewage sludge.**

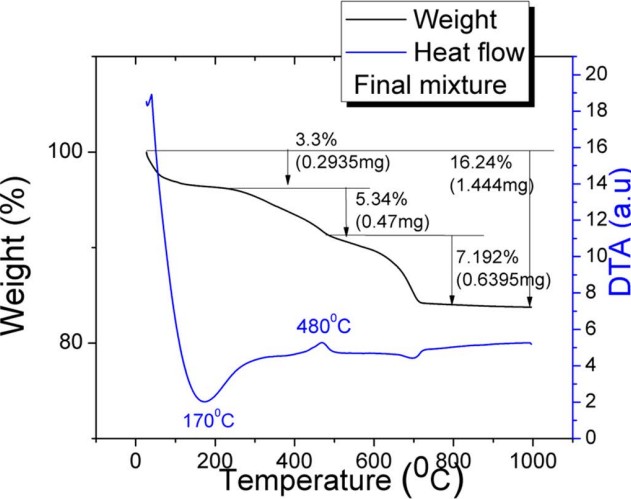

**Fig 9. TG for clays+ash +sewage sludge mixture (15%).**

### 3.3. Thermal parameters

A frequency scan of the normalized PPE phase is shown in Fig 10. The intersection of the experimental curve with X-axis ($f_0 = 13$ Hz) gives the thermal diffusivity of the sensor, according to the following expression: $\alpha_p = f_0 (L_p)^2 / \pi$.

The thermal diffusivity of the pyroelectric sensor is independent on the thermal behavior of the investigated sample. In this case, the sensor's thermal diffusivity is $\alpha = 1.1 \times 10^{-6}$ m²/s, in good agreement with the literature. This result confirms the fact that the theroretical and the experimental conditions are fulfilled. The obtained results for the investigated samples

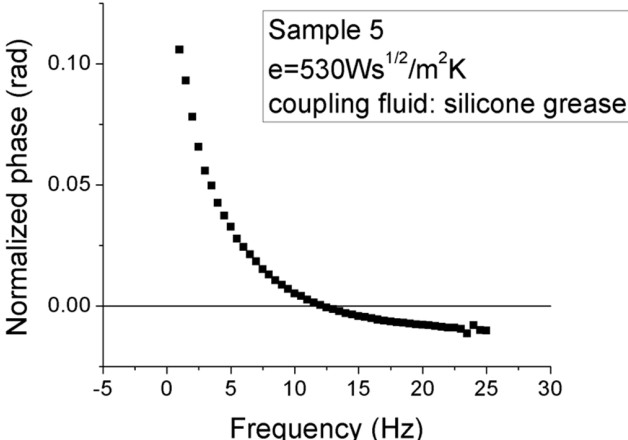

**Fig 10. Normalized FPPE phase as a function of chopping frequency for N20.**

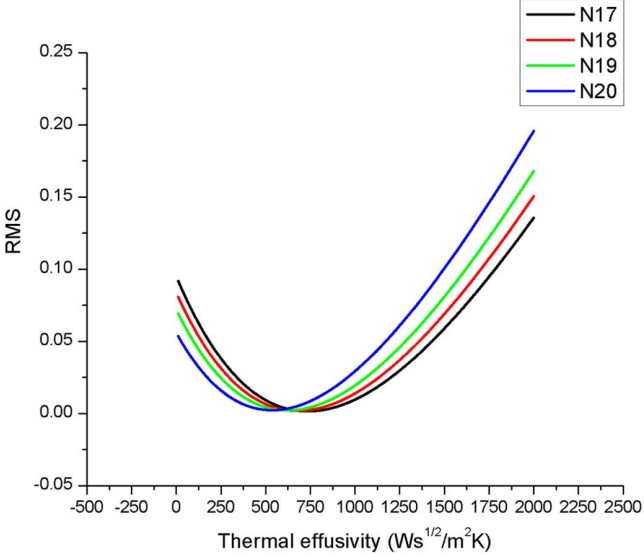

**Fig 11. Root mean square deviation of the fit performed with** Eq. (9) **on the experimental data (PPE phase as a function of chopping frequency), with sample's thermal effusivity as a fit parameter. The minimum of the curve indicates the value of the thermal effusivity.**

(N17 ÷ N20) are displayed in Fig 11. The figure shows the optimization of the fits performed in order to get the thermal effusivity. The correct value of the thermal effusivity minimizes the root mean square (RMS) of the experimental data performed with Eq. (9).

**IR investigations:** The DC image and its corresponding amplitude profile are shown in Fig 12, the amplitude image and its profile are presented in Fig 13, and the phase image along with its amplitude profile are illustrated in Fig 14. The continue and the amplitude images show the dissipation of the heat at the surface of the sample, due to the absorption of the laser. The profiles are quite smooth and thus the emissivity of the investigated surface can be assumed to be constant. The average temperature of the surface has increased from 26°C up to 41°C (see Fig 12). The profile of the phase image shows that the thermal wave diffuses to the surface of the sample symmetrically with respect to the excitation

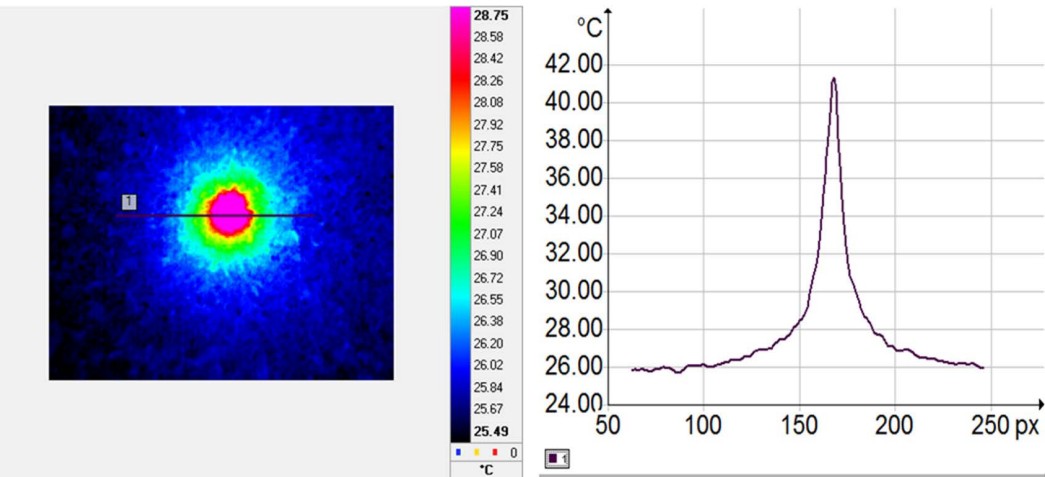

**Fig 12. DC image (a) and the amplitude profile along the marked line (b) for N20.**

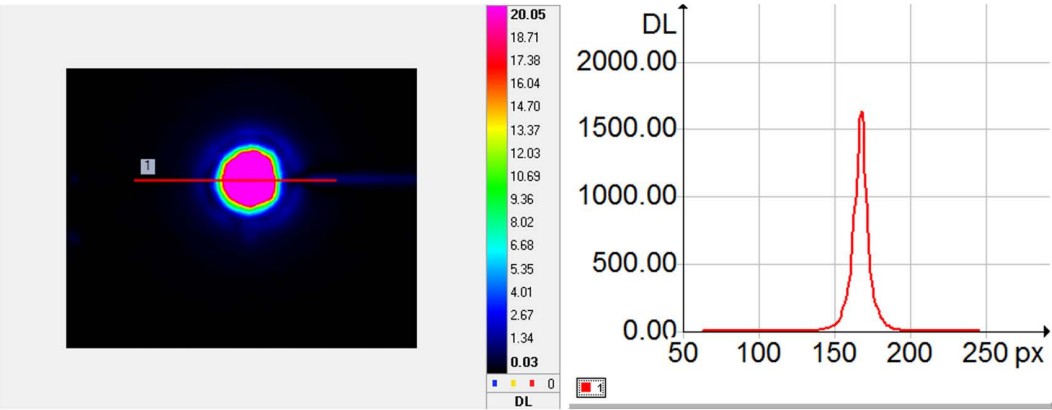

**Fig 13. Amplitude image (a) and the amplitude profile along the marked line (b) for N20.**

source (the laser is targeted close to the pixel 170 on the IR images). The thermal wave spreads over a distance of about 1.5 mm around the excitation source. The disturbance of the thermal wave (see the phase profile, around pixel 210) reveals the presence of a small pore located on the surface of the sample M2. The thermal diffusivity of the samples was calculated from phase profiles, according to Eq. 10.

The obtained results with the associated level of uncertainty are listed in Table 3. The thermal effusivity was measured for all four transversal sections of a cubic specimen and the average value was calculated. The thermal diffusivity was taken on the same surface, at four different points, and the mean value was calculated.

The thermal conductivity values lie between 0.7W/mK (the reference sample) and 0.51W/mK. The thermal conductivity of the samples containing dry sewage sludge pore-forming agent decreases by 30% compared to the reference sample.

Considering our results, the uncertainty in the thermal properties measured using lock-in thermography is quantified considering our experimental setup. The precision of the infrared camera, with a sensitivity of 20 mK, combined with phase shift measurements uncertainty of ±1%, and the uncertainty of the laser power and modulation frequency of ±1%, lead to an overall minimum uncertainty of ±2% for the thermal parameters. Since the uncertainties propagate differently

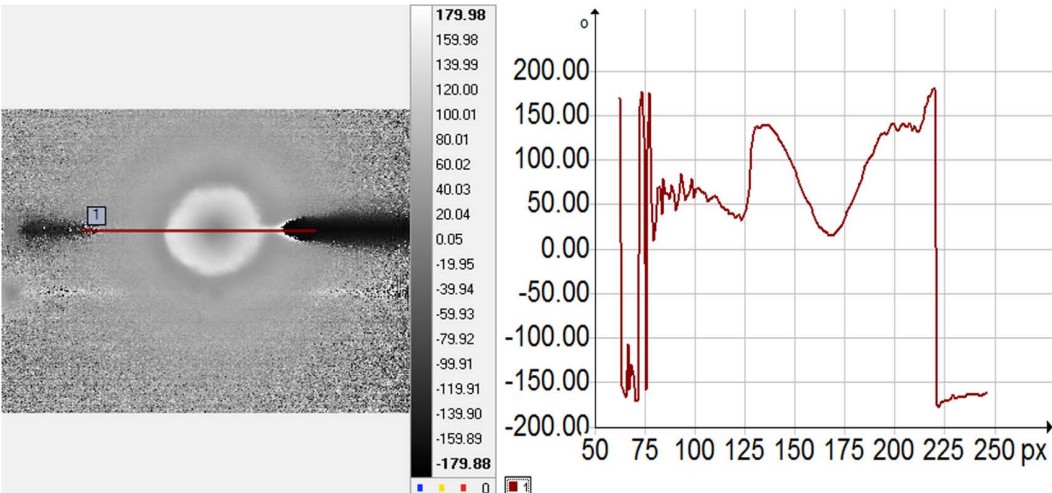

**Fig 14. Phase image (a) and the amplitude profile along the marked line (b) for N20.**

**Table 3. Thermal parameters.**

| Clay mix | α *10⁷m²/s | e (Ws^{1/2}/m²K) | k (W/mK) | Uncertainty (min to max) |
|---|---|---|---|---|
| Control brick | 9.2 | 730 | 0.70 | |
| N17 | 8.7 | 727 | 0.68 | ±2% to ±5% |
| N18 | 8.2 | 644 | 0.58 | |
| N19 | 7.9 | 622 | 0.54 | |
| N20 | 7.8 | 563 | 0.51 | |

for each parameter, the maximum uncertainty will vary depending on which parameter has the greatest impact. Considering also the contributions from the signal-to-noise ratio (SNR) together with an uncertainty increase due to the camera precision, the maximum uncertainty in the determination of thermal effusivity, conductivity, and diffusivity reaches up to approximately 5%.

## 5. Conclusions

This research successfully demonstrates the potential of using dry wastewater sludge as a pore-forming agent for large-scale production of fired clay bricks. The entire production process is characterized in terms of resulted bricks' physico-mechanical properties, thermal decomposition during burn in process and thermal parameters. Moreover, this study highlights the effectiveness of an innovative image processing approach that leverages computational design and engineering principles to accurately quantify porosity in ceramic materials, which is critical for optimizing their mechanical and thermal properties.

Regarding *physico-mechanical properties*, the incorporation of small pores within the ceramic structure has significantly enhanced the material's behavior. An increase in the percentage of small pores leads to a more uniform pore distribution, which prevents the formation of weak spots that could act as initiation points for cracks. As a result, the material's ability to withstand compressive forces of up to of 28 N/mm² for the 15% sewage sludge mixture is observed, ensuring that the ceramic blocks meet the compressive strength requirements for high-quality bricks. The hypothesis that sewage sludge can be effectively used as a pore-forming agent in ceramic production has been validated using the *thermal*

*decomposition* approach. Thus, this observation is in good agreement with the thermogravimetric analysis, confirming that the addition of sludge contributes to the desired porosity, enhancing the thermal properties of the ceramic material while aligning with the study's sustainability goals. Considering the *thermal parameters*, the results indicate that the studied samples exhibit low thermal conductivity (approximately 30% lower than the reference sample), which is essential for energy-efficient building applications. The incorporation of sewage sludge not only reduces the environmental impact of waste disposal but also provides a cost-effective and sustainable alternative for producing thermally efficient construction materials. The main findings suggest that while the incorporation of sludge enhances the thermal insulation properties of the bricks, it also leads to a decrease in compressive strength due to increased porosity. However, this trade-off is manageable and aligns with the goals of producing environmentally sustainable building materials.

Taking into account the *environmental impact and future applications*, the reuse of dry sewage sludge as a pore-forming agent in fired bricks presents a promising solution for waste valorization, significantly reducing environmental pollution while conserving natural resources. The integration of advanced image processing techniques and thorough material characterization ensures that the resulting bricks meet the current standards of efficiency and performance in modern construction. In summary, this study underscores the potential of innovative material processing techniques to address critical challenges in sustainable construction, offering a robust framework for future research and industrial applications. The findings encourage further exploration into optimizing the balance between mechanical strength and thermal efficiency, with the ultimate goal of advancing the production of environmentally friendly building materials.

## Acknowledgments

The research was funded by a grant from the Romanian Ministry of Education and Research, UEFISCDI, under project no. 61TE, code PN-IV-P2-2.1-TE-2023–0300, and by the Experimental-Demonstrative Project PN-IV-P7-7.1-PED-2024–2349, grant number 32PED/08/01/2025, project acronim MLCELLM.

## Author contributions

**Conceptualization:** Aurel Mihail Baloi, Mihaela Streza, Belean Bogdan.

**Data curation:** Aurel Mihail Baloi.

**Investigation:** Aurel Mihail Baloi.

**Methodology:** Mihaela Streza, Belean Bogdan.

**Software:** Aurel Mihail Baloi, Belean Bogdan.

**Supervision:** Mihaela Streza.

**Validation:** Aurel Mihail Baloi, Mihaela Streza, Belean Bogdan.

**Visualization:** Aurel Mihail Baloi, Belean Bogdan.

**Writing – original draft:** Aurel Mihail Baloi, Belean Bogdan.

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
