## [Decision Letter · Decision Letter 0]

16 Apr 2025

PONE-D-25-05897Light Microscopy Image Segmentation Using Active Contours Driven by Local Image Information for Environmentally Friendly Fired-clay Bricks Design and CharacterizationPLOS ONE

Dear Dr. Bogdan,

Thank you for submitting your manuscript to PLOS ONE. After careful consideration, we feel that it has merit but does not fully meet PLOS ONE’s publication criteria as it currently stands. Therefore, we invite you to submit a revised version of the manuscript that addresses the points raised during the review process.

Please comply with the reviewers' request.

We look forward to receiving your revised manuscript.

Kind regards,

André Gustavo de Sousa Galdino

Academic Editor

PLOS ONE

 [TE2024, PED2024]. 

Reviewers' comments:

Reviewer's Responses to Questions

**Comments to the Author**

1. Is the manuscript technically sound, and do the data support the conclusions?

Reviewer #1: Partly

Reviewer #2: Yes

Reviewer #3: Yes

2. Has the statistical analysis been performed appropriately and rigorously? 

Reviewer #1: No

Reviewer #2: N/A

Reviewer #3: Yes

3. Have the authors made all data underlying the findings in their manuscript fully available?

Reviewer #1: Yes

Reviewer #2: Yes

Reviewer #3: Yes

4. Is the manuscript presented in an intelligible fashion and written in standard English?

Reviewer #1: No

Reviewer #2: Yes

Reviewer #3: Yes

5. Review Comments to the Author

Reviewer #1: The manuscript presents an experimental study on fired clay bricks incorporating sewage sludges. The manuscript is not written well and is not well-organized. The novelty of the research is not clearly demonstrated. There are many relevant studies on the same topic on using sewage sludges in fired clay bricks (e.g., https://doi.org/10.1063/1.5002344;
https://doi.org/10.1016/j.rineng.2023.101708). The authors claimed that incorporating the light microscopy image in analyzing the porosity of clay bricks in the manuscript. However, there is a noticeable difference in porosity measured between MIP and proposed image processing method, as shown in Table 2. The relevant statistical approach should be also incorporated to highlight the advantage of the proposed image processing method. Language editing is recommended as there are multiple typos/grammar issues in the manuscript. The specific comments, questions, and suggestions are shown below:

1. The Introduction section is too long and lacks clarity and logical flow. It should be more directly connected to the research topic rather than presenting excessive general information (e.g., lines 160-171). Additionally, Section 1.2 should be removed and its key points integrated into a concise paragraph that effectively highlights the significance of this study.

2. In section 2.1, the material characterization tests (e.g., XRD, TGA, XRF, etc.) for raw materials (clays and dry sewage sludge) should be performed.

3. Line 291: the pressure unit is incorrect. There are lots of similar typos throughout the manuscript.

4. Line 295: Cooling rate is suggested to incorporate.

5. In section 2.2, the testing standards and references should be given. The authors did not present all tests performed in this study, e.g., water absorption.

6. Line 544: wather?

7. Lines 590-603: Degree signs are misused.

8. Although the 15% sewage sludge mixture achieved a compressive strength of 28 MPa (Table 1), exceeding the ASTM C62 requirements for brick strength, its water absorption increased to 13.7%. This rise in water absorption could significantly compromise the bricks' freeze-thaw durability. How did the authors address or mitigate this issue?

Reviewer #2: The manuscript presents several positive aspects that contribute to its value in the field of materials science and sustainable construction:

- It addresses not only waste management issues but also promotes the development of eco-friendly construction materials;

- The paper proposes an advanced image segmentation method using active contours driven by local image information. This technique effectively estimates porosity in ceramic brick mixtures, overcoming challenges such as light reflections and intensity inhomogeneity, which are common in microscopy;

- employs a comprehensive methodology that includes evaluations of thermal and mechanical properties. The use of non-contact infrared lock-in thermography and thermogravimetric analysis (TGA) provides accurate assessments of the material's thermal response and mineral weight loss during decomposition;

- The paper emphasizes the importance of producing sustainable and economical construction materials. By incorporating sewage sludge as a pore-forming agent, the study highlights a practical solution to improve the thermal properties of ceramic bricks while addressing environmental issues;

- It is well-structured, with a logical flow from the introduction to the methodology and results;

- The results indicate that the incorporation of sewage sludge increases porosity, which enhances the insulating properties of the bricks. Notably, the compressive strength of mixtures containing 15% sewage sludge meets the criteria for first-class bricks according to ASTM C62 standards, demonstrating the viability of the proposed method.

Although the manuscript presents several positive contributions, there are some aspects that could be improved:

- The proposed image segmentation method, although innovative, may be complex for practical applications. The reliance on active contours driven by local image information may require specialized knowledge and software, which could limit its accessibility for broader use in the industry;

- While thermal properties are enhanced, the addition of sewage sludge may reduce the mechanical strength of the bricks, posing challenges for structural applications, as the compressive strength may not meet all construction standards;

- The manuscript does not address the potential variability in results due to different proportions of sewage sludge or other materials. A more thorough exploration of how varying these proportions affects thermal and mechanical properties could provide valuable insights for manufacturers;

- The study could be improved by including a comparative analysis with existing methods for producing fired clay bricks. This would help contextualize the benefits of the proposed approach and demonstrate its advantages over traditional methods;

- Expanding the tests to include a wider range of conditions and material combinations would strengthen the conclusions and applicability of the research.

Reviewer #3: 1- While the proposed active contour approach with local image information is novel, it is crucial to clearly justify why traditional approaches (such as standard Chan-Vese or machine learning techniques) are insufficient specifically for the porosity characterization of ceramic bricks. A detailed comparative analysis highlighting specific limitations would strengthen the methodological justification.

2- The paper lacks an explicit quantitative validation method to rigorously demonstrate the accuracy of the proposed segmentation technique. Introducing performance metrics and clearly comparing the proposed approach with established methods would enhance credibility.

3- The authors discussed addressing the defocus effect using the Laplace operator. However, no clear experimental results quantitatively demonstrate the effectiveness of this approach. Include detailed experimental data or quantitative metrics that explicitly demonstrate how effectively the proposed method handles defocus compared to existing methods.

4- The manuscript indicates enhanced porosity leading to improved insulation properties; however, little attention is paid to the microstructural homogeneity of pores. Additional microstructural analyses, such as SEM images, would complement the existing microscopic data and confirm that pore formation remains homogeneous across samples.

5- Although the study demonstrates acceptable compressive strength for bricks incorporating up to 15% sewage sludge, a clear quantitative relationship or regression analysis linking porosity levels (quantitatively segmented via microscopy) to the observed mechanical strength would strengthen the practical relevance of the findings.

6- The thermogravimetric analysis provided valuable insights; however, the manuscript does not sufficiently correlate TGA mass loss results with observed changes in porosity or thermal conductivity values. Clearly establishing these relationships would add depth to the discussion on thermal decomposition dynamics in brick manufacturing.

7- Stability and reproducibility of the segmentation results were not explicitly discussed. It would be beneficial if the authors performed a repeatability analysis by processing multiple images under slightly varied lighting or focus conditions to verify the robustness of the proposed segmentation method.

8- While MIP is cited as the comparative standard, the manuscript briefly acknowledges its limitations without clearly stating how this could affect comparative assessments. Clarify and explicitly state how these limitations were considered or compensated for during comparisons between MIP and the proposed image-based method.

9- For thermal property measurements using non-contact infrared lock-in thermography, it is important to state explicitly the measurement uncertainty and any assumptions in thermal diffusivity and conductivity calculations. Including these details will enhance the reliability of the thermal characterization results.

10- Given the scope and content of this paper, it may benefit from considering the following related works:

https://doi.org/10.3390/su15129389

https://doi.org/10.1088/1755-1315/899/1/012042

https://doi.org/10.1061/(ASCE)MT.1943-5533.0003737

6. PLOS authors have the option to publish the peer review history of their article (what does this mean? ). If published, this will include your full peer review and any attached files.

**Do you want your identity to be public for this peer review?** For information about this choice, including consent withdrawal, please see our Privacy Policy .

Reviewer #1: No

Reviewer #2: No

Reviewer #3: No

---

## [Author Response · Author response to Decision Letter 1]

16 May 2025

We are also grateful for the reviewers’ thoughtful evaluations, which significantly contributed to improving the manuscript. We have addressed all the comments and suggestions provided by the editors and reviewers, as detailed in the response to reviewers' letter. All corresponding modifications in the manuscript are highlighted in yellow.

---

## [Decision Letter · Decision Letter 1]

29 Jun 2025

Light Microscopy Image Segmentation Using Active Contours Driven by Local Image Information for Environmentally Friendly Fired-clay Bricks Design and Characterization

PONE-D-25-05897R1

Dear Dr. Bogdan,

We’re pleased to inform you that your manuscript has been judged scientifically suitable for publication and will be formally accepted for publication once it meets all outstanding technical requirements.

Kind regards,

Hailing Ma

Academic Editor

PLOS ONE

Additional Editor Comments (optional):

Reviewers' comments:

Reviewer's Responses to Questions

**Comments to the Author**

1. If the authors have adequately addressed your comments raised in a previous round of review and you feel that this manuscript is now acceptable for publication, you may indicate that here to bypass the “Comments to the Author” section, enter your conflict of interest statement in the “Confidential to Editor” section, and submit your "Accept" recommendation.

Reviewer #1: All comments have been addressed

Reviewer #3: (No Response)

2. Is the manuscript technically sound, and do the data support the conclusions?

Reviewer #1: Yes

Reviewer #3: (No Response)

3. Has the statistical analysis been performed appropriately and rigorously? 

Reviewer #1: N/A

Reviewer #3: (No Response)

4. Have the authors made all data underlying the findings in their manuscript fully available?

Reviewer #1: Yes

Reviewer #3: (No Response)

5. Is the manuscript presented in an intelligible fashion and written in standard English?

Reviewer #1: Yes

Reviewer #3: (No Response)

6. Review Comments to the Author

Reviewer #1: (No Response)

Reviewer #3: Upon reviewing the authors' responses and the revised manuscript, the paper has been significantly improved and now meets the publication criteria of the journal. Acceptance of the manuscript is recommended.

7. PLOS authors have the option to publish the peer review history of their article (what does this mean? ). If published, this will include your full peer review and any attached files.

**Do you want your identity to be public for this peer review?** For information about this choice, including consent withdrawal, please see our Privacy Policy .

Reviewer #1: No

Reviewer #3: No

---

## [Editor Report · Acceptance letter]

PONE-D-25-05897R1

PLOS ONE

Dear Dr. Bogdan,

I'm pleased to inform you that your manuscript has been deemed suitable for publication in PLOS ONE. Congratulations! Your manuscript is now being handed over to our production team.

Kind regards,

on behalf of

Dr. Hailing Ma

Academic Editor

PLOS ONE